# Impact of Pre-Existing History of Heart Failure on Patient Profile, Therapeutic Management, and Prognosis in Cardiogenic Shock: Insights from the FRENSHOCK Registry

**DOI:** 10.3390/life12111844

**Published:** 2022-11-11

**Authors:** Guillaume Schurtz, Clément Delmas, Margaux Fenouillet, François Roubille, Etienne Puymirat, Laurent Bonello, Guillaume Leurent, Basile Verdier, Bruno Levy, Julien Ternacle, Brahim Harbaoui, Gerald Vanzetto, Nicolas Combaret, Benoît Lattuca, Cedric Bruel, Jeremy Bourenne, Vincent Labbé, Patrick Henry, Éric Bonnefoy-Cudraz, Nicolas Lamblin, Gilles Lemesle

**Affiliations:** 1USIC et Centre Hémodynamique, Institut Coeur Poumon, Centre Hospitalier Universitaire de Lille, 59000 Lille, France; 2Intensive Cardiac Care Unit, Rangueil University Hospital/Institute of Metabolic and Cardiovascular Diseases (I2MC), UMR-1048, National Institute of Health and Medical Research (INSERM), 1 Avenue Jean Poulhes, 31059 Toulouse, France; 3Cardiology Department, INI-CRT, CHU de Montpellier, PhyMedExp, Université de Montpellier, INSERM, CNRS, 34000 Montpellier, France; 4Department of Cardiology, Assistance Publique des Hôpitaux de Paris, 75000 Paris, France; 5Cardiology Department, APHM, Mediterranean Association for Research and Studies in Cardiology (MARS Cardio), Centre for CardioVascular and Nutrition Research (C2VN), Aix-Marseille University, INSERM 1263, INRA 1260, 13000 Marseille, France; 6Department of Cardiology, CHU Rennes, Inserm, LTSI-UMR 1099, University Rennes 1, 35000 Rennes, France; 7Service de Réanimation Médicale Brabois, CHRU Nancy, Pôle Cardio-Médico-Chirurgical, INSERM U1116, Faculté de Médecine, Vandoeuvre-les-Nancy, Université de Lorraine, 54000 Nancy, France; 8Hôpital Cardiologique Haut-Lévêque, Centre Hospitalier Universitaire de Bordeaux, 33318 Pessac, France; 9Cardiology Department, Hôpital Croix-Rousse and Hôpital Lyon Sud, Hospices Civils de Lyon, 69000 Lyon, France; 10CREATIS UMR5220, INSERM U1044, INSA-15, University of Lyon, 69000 Lyon, France; 11Department of Cardiology, Hôpital de Grenoble, 38000 Grenoble, France; 12Department of Cardiology, CHU Clermont-Ferrand, CNRS, Université Clermont Auvergne, 63000 Clermont-Ferrand, France; 13Department of Cardiology, Nîmes University Hospital, Montpellier University, 30000 Nîmes, France; 14Medical and Surgical Intensive Care Unit, Groupe Hospitalier Paris Saint Joseph, 75000 Paris, France; 15Service de Réanimation des Urgences, CHU La Timone 2, Aix Marseille Université, 13000 Marseille, France; 16Medical Intensive Care Unit, AP-HP, Tenon University Hospital, 75000 Paris, France; 17Department of Cardiology, Assistance Publique-Hôpitaux de Paris, INSERM U942, University of Paris, 75000 Paris, France; 18Intensive Cardiological Care Division, Hospices Civils de Lyon-Hôpital Cardiovasculaire et Pulmonaire, 69000 Lyon, France; 19Cardiology Department, Heart and Lung Institute, University Hospital of Lille, 59000 Lille, France; 20INSERM U1167, Institut Pasteur of Lille, 59000 Lille, France; 21Heart and Lung Institute, University Hospital of Lille, 59000 Lille, France; 22Inserm U1011, Institut Pasteur of Lille, 59000 Lille, France; 23FACT (French Alliance for Cardiovascular Trials), 75000 Paris, France

**Keywords:** cardiogenic shock, heart failure, myocardial infarction, mechanical circulatory support

## Abstract

There is a large heterogeneity among patients presenting with cardiogenic shock (CS). It is crucial to better apprehend this heterogeneity in order to adapt treatments and improve prognoses in these severe patients. Notably, the presence (or absence) of a pre-existing history of chronic heart failure (CHF) at time of CS onset may be a significant part of this heterogeneity, and data focusing on this aspect are lacking. We aimed to compare CS patients with new-onset HF to those with worsening CHF in the multicenter FRENSHOCK registry. Altogether, 772 CS patients were prospectively included: 433 with a previous history of CHF and 339 without. Worsening CHF patients were older (68 +/− 13.4 vs. 62.7 +/− 16.2, *p* < 0.001) and had a greater burden of extra-cardiac comorbidities. At admission, acute myocardial infarction was predominantly observed in the new-onset HF group (49.9% vs. 25.6%, *p* < 0.001). When focusing on hemodynamic parameters, worsening CHF patients showed more congestion and higher ventricular filling pressures. Worsening CHF patients experienced higher in-hospital all-cause mortality (31.3% vs. 24.2%, *p* = 0.029). Our results emphasize the great heterogeneity of the patients presenting with CS. Worsening CHF patients had higher risk profiles, and this translated to a 30% increase in in-hospital all-cause mortality. The heterogeneity of this population prompts us to better determine the phenotype of CS patients to adapt their management.

## 1. Introduction

Acute heart failure (AHF) remains a major cause of hospitalization and is burdened by a consequent high morbidity, high mortality, and a high rate of re-hospitalizations [1]. Cardiogenic shock (CS) represents the most severe form of AHF and is still associated with a very high rate of mortality (up to 60–70% at 1 year) [2,3,4]. It accounts for 2 to 5% of AHF patients [5,6], and its prevalence in intensive care units (ICU) can reach 14–16% of the total admissions [5,7].

Notably, there is a large diversity among patients presenting with CS, which may influence the hemodynamic profile, patient management, and prognosis. The different causes (underlying cardiopathy) that lead to CS is a significant part of this heterogeneity. Historically, the predominant cause of CS was ischemic (up to 80%), and between 5 and 10% of myocardial infarctions (MIs) were complicated by CS [8,9,10,11,12]. However, there is now a trend towards a decrease in ischemic etiologies [13,14]. For example, non-ischemic causes were recently identified as predominant causes of CS in the FRENSHOCK registry [13]. Furthermore, beyond the manifold etiologies, the presence or absence of a pre-existing documented cardiopathy and history of heart failure (HF) at the time of CS occurrence may further accentuate the heterogeneity of these patients and therefore their optimal management and prognosis. Indeed, CS may be the initial presentation of HF, occurring on a previously “healthy” heart or revealing an unknown cardiopathy. Alternatively, it may be the umpteenth destabilization of a well-documented chronic cardiopathy that is a fortiori treated and followed.

So far, there are very few data about the potential differences in patient profiles, shock management, and prognosis among these two distinct populations. In this post hoc analysis of the largest European registry of CS (the FRENSHOCK multicenter registry), we thus attempted to compare CS patients according to the presence or absence of a pre-existing history of HF at the time of admission.

## 2. Methods

### 2.1. Patient Population

The design of the FRENSCHOCK registry has already been published in detail [13,15]. Briefly, FRENSHOCK is a prospective multicenter registry conducted over a 6-month period in France between April and October 2016 (NCT02703038).

All patients (n = 772) presenting with CS were included from 52 recruiting centers if they met at least one criterion of each of the following: (i) hemodynamic criteria, defined as a low systolic blood pressure (SBP) < 90 mmHg and/or the need for maintenance with vasopressors/inotropes and/or a low cardiac index < 2.2 L/min/m^2^; (ii) left and/or right heart pressure elevation, defined by clinical signs, radiology, blood tests, echocardiography, or signs of invasive hemodynamic overload; and (iii) signs of organ malperfusion, which could be either clinical and/or biological. Patients could be included regardless of CS etiology and whether CS was present at admission or developed during their in-hospital course. Exclusion criteria were refusal to participate, shock from a non-cardiac origin, and post-cardiotomy CS.

The study was conducted in accordance with the guidelines for good clinical practice and French law. Written consent was obtained from all the patients. The data recorded and their handling and storage were reviewed and approved by the CCTIRS (French Health Research Data Processing Advisory Committee) (n° 15.897) and the CNIL (French Data Protection Agency) (n° DR-2016-109).

### 2.2. Objectives and Outcomes

The main objective of the present post hoc analysis was to assess the impact of a pre-existing history of HF on CS management and clinical outcomes. We thus compared patients with a previous diagnosis of HF (n = 433) to those without (n = 339). New-onset HF was defined as the patient having no previous history of HF after an interrogation of the patient, the patient’s family, and/or the usual practitioner. Patients with a history of HF were classified as having worsening chronic HF (CHF).

The primary outcome was in-hospital all-cause mortality.

### 2.3. Funding and Data Property

This registry emanated from the French Society of Cardiology and was endorsed by its Emergency and Acute Cardiovascular Care Working Group. The study was sponsored by the “Fédération Francaise de Cardiologie” and was funded by unrestricted grants from Daiichi-Sankyo and Maquet SAS.

### 2.4. Statistical Analysis

Continuous variables are reported as means +/− standard deviation (SD) or medians and interquartile ranges (IQR) as appropriate. Discrete variables are described as absolute numbers and percentages.

Groups (pre-existing history of CHF or not) were compared using Student’s *t* test or ANOVA for continuous variables and χ^2^ or Fisher’s exact test for discrete variables as appropriate.

To determine independent predictors of in-hospital all-cause mortality, a multivariate logistic regression analysis was used. Variables included in the final models were selected ad hoc based on their physiological relevance and potential to be associated with outcomes. Included variables were: group (pre-existing history of CHF or not), age, gender, systolic blood pressure (SBP) at admission, heart rate at admission, left ventricle ejection fraction (LVEF) at admission, presentation as MI, creatinine level at admission, arterial lactate level at admission, hemoglobin level at admission, and prothrombin time (PT) level at admission. Odds ratios (oRs) are presented with 95% confidence intervals (cIs).

Statistical analyses were performed using IBM SPSS 23.0 (IBM SPSS Inc., Chicago, IL, USA). For all analyses, two-sided *p* values < 0.05 were considered significant.

## 3. Results

### 3.1. Baseline Characteristics

Baseline characteristics of the 772 included patients are summarized in Table 1. The mean age was 65.7 (+/− 14.9) and 71.5% of the patients were men. Histories of diabetes mellitus were present in 28% of the patients. History of renal failure and history of chronic obstructive pulmonary disease (COPD) or chronic respiratory failure were present in 29.9% and 7% of the patients, respectively.

Among the 433 patients admitted with worsening CHF (56.1%), ischemic etiology accounted for 231 patients (29.9%). Others’ causes were represented by idiopathic dilated cardiomyopathy (n = 89), left-sided valvular stenosis or incompetence (n = 81), hypertrophic cardiomyopathy (n = 13), toxic causes (n = 35), and others (n = 69). Altogether, 85 (11%) patients had mixed causes.

Patients with worsening CHF were older than those with new-onset HF (68 +/− 13.4 vs. 62.7 +/− 16.2, *p* < 0.001), and there were more men (76.2% vs. 65.5%, *p* = 0.001). In addition, more of these patients had a history of diabetes (33.3% vs. 21.5%, *p* < 0.001), renal failure (32.3% vs. 7.1%, *p* < 0.001), and respiratory failure (8.8% vs. 4.7%, *p* = 0.029).

### 3.2. Initial Presentation

These data are depicted in Table 2. In the overall population, the mean heart rate was 96 +/− 30 beats per minute and the mean SBP was 101 +/− 25 mmHg. Only 51.9% of the patient presented with sinus rhythm. The mean LVEF was 26.3% +/− 13.4, and 72.2% of the patients had a LVEF below 30%. Acute MI was identified as the precipitating factor of CS occurrence in 36.3% of the cases and severe sepsis in 11.9%.

Patients with worsening CHF had a lower heart rate than those with new-onset HF (91 +/− 28 vs. 100 +/− 31, *p* < 0.001) and presented with sinus rhythm less often (46.2% vs. 59.4%, *p* < 0.001). They also had lower LVEFs (25% +/− 12.2 vs. 28% +/− 14.5, *p* = 0.003). Acute MI was less frequently the precipitating factor in patients with worsening CHF (25.6% vs. 49.9%, *p* < 0.001) (Figure 1).

### 3.3. Biological Parameters

Biology parameters are shown in Table 3. In the overall study population, the median estimated glomerular filtration rate (eGFR) was 46 (28–67) mL/min. The median hemoglobin was 12.6 (11–14) g/dL. The median arterial lactate level was 3 (2–4.75) mmol/L, and 61.7% of the patients had a lactate level above the normal value. The median bilirubin level was 16 (9–29) mg/L and the median PT was 59% (37–77). The median BNP and NT-pro-BNP were 1150 (476–2778) and 9277 (4045–23,810) pg/mL, respectively. The median CRP level was 28 (9–69) mg/L.

Worsening CHF patients had significantly lower eGFRs (40 (25–59) vs. 56 (39–77), *p* < 0.001) and hemoglobin levels (12 (10.6–14) vs. 13 (11.2–14.6), *p* < 0.001) than those with new-onset HF. They also had higher bilirubin (20 (11–34) vs. 13 (8–23), *p* < 0.001) and lower PT (48 (29–68) vs. 70 (52–85), *p* < 0.001). When assessed by deciles, BNP/NT-pro-BNP was significantly higher in patients with worsening CHF (*p* < 0.001).

### 3.4. Right Heart Catheterization Parameters

In our population, 83 patients had right heart catheterization: 36 in the new-onset HF group and 47 in the worsening CHF group (Table 4). The median right atrial pressure (RAP) was 10 (6–14) mmHg, the median mean pulmonary artery pressure (mPAP) was 29 (24–35) mmHg, and the median pulmonary capillary wedge pressure (PCWP) was 19 (14–25). The median cardiac index (CI) was 2.1 (1.9–3) L/min/m^2^.

Patients of the worsening CHF group had higher RAP (12 (6–16) vs. 8 (5–11), *p* = 0.028) and a trend for higher PCWP (22 (15–29) vs. 16 (13–23), *p* = 0.062).

### 3.5. Shock Management

In-hospital management is shown in Table 5. In the overall population, vasoactive drugs were prescribed as follows: dobutamine in 82.2% of the patients for a median time of 5 (2–8) days, noradrenaline in 53.3% of the patients for a median time of 3 (2–6) days, adrenaline in 12.3% of the patients for a median time of 1 (1,2) day, and levosimendan in only 7.4% of the cases. Intravenous diuretics were used in 82% of the cases and volume expansion in 41.6%. Organ replacement therapies were used as follows: mechanical ventilation (MV) in 291 patients (37.9%), renal replacement therapy (RRT) in 122 patients (15.8%), and mechanical circulatory support (MCS) in 143 patients (18.6%), including 48 patients with an intra-aortic balloon pump (IABP) for a median time of 2 (1–5) days, 26 patients with an Impella^®^ device for a median of 6 (3–9) days, and 85 patients with extracorporeal life support (ECLS) for a median time of 5 (3–8) days. Coronary angiographies were performed on 399 patients (51.7%) and culprit lesions were treated in 256 patients (64.2% of those who had undergone a coronary angiography).

Noradrenaline and adrenaline were used less often in patients presenting with worsening CHF (49.9% vs. 57.9%, *p* = 0.03 and 10.2% vs. 15.1%, *p* = 0.04, respectively). Diuretics were used more often (86.8% vs. 76.9%, *p* < 0.001) and volume expansion was used less often (35.6% vs. 49.9%, *p* < 0.001). Organ replacement therapies, i.e., MV (26.9% vs. 51.8%, *p* < 0.001) and MCS (13% vs. 25.6%, *p* < 0.001) were also used less often. Coronary angiographies were performed less frequently (38.3% vs. 68.7%, *p* < 0.001) and culprit lesions were less frequently identified (51.2% of those who undergone a coronary angiography vs. 73.4%, *p* < 0.001).

### 3.6. In-Hospital Outcomes

The median in-hospital length of stay was 16 (10–26) days and the median length of stay in ICU was 11 (7–21) days. The in-hospital all-cause mortality rate was 28% (n = 217 patients).

Worsening CHF patients had a non-significant trend for longer in-hospital stays (16 (10–27) days vs. 14 (9–24) days, *p* = 0.056). They experienced a higher all-cause mortality rate (31.3% vs. 24.2%, *p* = 0.029, Figure 2). After multivariate analysis, independent predictors of in-hospital all-cause mortality were age, heart rate, SBP, LVEF, hemoglobin, and arterial blood lactate levels (Table 6). Worsening CHF was not independently associated with in-hospital all-cause mortality (OR = 1.051, 95% CI = 0.680–1.624, *p* = 0.823).

## 4. Discussion

AHF and especially patients presenting with CS continue to experience poor in-hospital prognosis [8,11,12]. Incremental insights into risk stratification could be of paramount importance to improve patient management and therefore clinical outcomes. CS related to acute MI (AMI-CS) has been the focus of intense investigations [16,17]. However, the number of CS patients related to other etiologies (commonly designated as HF-CS) is now more prevalent in the contemporary era [7,13]. Clinical characteristics and in-hospital outcomes between HF-CS and AMI-CS were recently reported and showed better survival in HF-CS patients [18]. One could argue that AMI patients often represent the vast majority of new-onset HF etiologies among CS patients and are therefore equal to new-onset HF. This assumption was not observed in our population and is probably unfounded. Indeed, CS was the first manifestation of HF for almost half of our patients and among them, AMI was considered as the triggering factor in only 54.8% of the cases. Interestingly, AMI was also identified as a precipitating factor for 31.6% of the worsening CHF patients. The whole spectrum of CS patients is clearly difficult to apprehend and there is a large variability among patient profiles in daily practice. In our study, worsening CHF patients had more extra-cardiac comorbidities as compared to those of the new-onset HF group. They were also significantly older, had more history of diabetes, renal failure, COPD, and peripheral artery disease. This observation is in perfect accordance with the previous literature in the field [19,20].

Notably, the CS pathophysiology of new-onset HF or decompensation of a pre-existing CHF may considerably differ. Indeed, new-onset HF is more often characterized by a sudden decrease in cardiac output without adaptative remodeling, resulting in a more pronounced hypoperfusion; in comparison, CHF is progressive by nature, with prolonged exposure to neurohormonal and hemodynamic perturbations that will durably affect all organs (especially the liver and kidneys) and lead to histological and functional changes even in the absence of severe tissular hypoperfusion and/or hypoxia. Different phenotypes of CS according to HF duration before the index event are therefore not surprising. The CS hemodynamic profile at presentation is of major importance when it comes to treatment adaptation and prognosis evaluation. Recently, Zweck et al. described three distinct phenotypes using a machine-learning approach (demographic, hemodynamic, and biological variables were used): non-congested (phenotype I), cardio-renal (phenotype II), and cardio-metabolic CS (phenotype III), which were closely linked to in-hospital deaths, the worst prognosis being observed for the phenotype III [21]. Importantly, mortality trends of these three phenotypes were not significantly different in AMI-CS and HF-CS patients (respectively, 21% vs. 10% for phenotype I, 45% vs. 32% for phenotype II, and 55% vs. 52% for phenotype III). In addition, Thayer et al. yielded important insights and showed a close association between CS mortality and patients’ congestion profiles based on the evaluation of left and right ventricular pressures [18]. They stratified patients into four categories (euvolemic, left, right, or bi-ventricular congestion). Right-sided and bi-ventricular congestion were significantly associated with the highest risks of in-hospital mortality in both AMI-CS and HF-CS subgroups. In multivariate analysis, the CS profile but not the cause of CS (AMI vs. not) was independently associated with mortality in their study. These studies have highlighted the fact that CS mortality seems to be related to patient hemo-metabolic status rather than to the underlying etiology of CS, even if both are sometimes strongly correlated. All these concerns may have a critical impact on therapeutic management.

In our analysis, worsening CHF patients showed a higher degree of congestion with higher RAP and a trend for higher PCWP, which seems consistent with initial CS management: more intravenous diuretic use and less volume expansion. Management of CS also involves hemodynamic stabilization with catecholamine therapy and sometimes organ support (MV, RRT, MCS). Despite limited available data on its true efficacy in improving prognosis, use of MCS is growing in the contemporary era. MCS is, however, associated with a high rate of complications that could subsequently influence outcomes, including major bleeding, thrombosis, hemolysis, stroke, and infection. MCS was inserted in 18.6% of the patients in our population, with a more frequent use in new-onset HF patients (25.6% vs. 13%, *p* = 0.001), most likely related to the fact that these patients were younger and had less extra-cardiac comorbidities. Interestingly, the duration of HF has an important impact on prognosis in patients receiving MCS. In the INTERMACS registry, despite a more severe presentation at the time of MCS implantation, acute de novo HF patients had better prognoses than CHF ones in each level of INTERMACS severity [22].

Among patients presenting with AHF, previous studies have revealed that pre-existing history of HF independently predicts mortality [23,24]. Data emanating from the ASCEND-HF trial revealed that HF duration ≥ 1 month was associated with increased mortality (>1 to 12 months, hazard ratio (HR): 1.89; 95% CI: 1.35 to 2.65; >12 to 60 months, HR: 1.82; 95% CI: 1.33 to 2.48; and >60 months, HR: 2.02; 95% CI: 1.47 to 2.77) [24]. These results were in accordance with those of a Danish nationwide cohort [23], which included 17,176 patients with a first admission for AHF. In this cohort, worsening CHF patients (n = 8316) had a higher rate of all-cause mortality or HF readmission during follow-up (HR 1.37, 95% CI 1.31–1.43). Moreover, it has been reported that a shorter duration of HF was associated with a higher probability of function recovery and a better prognosis in several clinical scenarios, such as before cardiac resynchronization therapy implantation [25,26]. However, these studies excluded patients with CS. Data on this specific subset of patients are sparse and represent an important gap in knowledge. In our large prospective cohort of CS patients, the worsening CHF subgroup was associated with a greater burden of extra-cardiac comorbidities, lower LVEF, and more pronounced organ failure. Importantly, in-hospital mortality was 30% higher in the worsening CHF group as compared to the new-onset HF group (31.3% vs. 24.2%). A pre-existing history of HF, however, was not independently associated with in-hospital mortality by multivariate analysis, highlighting again that these patients largely differ in many points such as age, extra-cardiac comorbidities, and hemodynamic profile.

## 5. Strengths and Limitations

Our work prospectively included 772 consecutive CS patients (the larger European cohort) from a broad spectrum of etiologies. In addition, we used a contemporary and pragmatic definition of CS that considerably strengthens our results. However, it may be challenging in clinical daily practice to determine the chronicity of HF, and some patients could have been misclassified. Initial admission in primary centers may have influenced early patients’ management and may explain the high rate of IABP use. Few patients underwent right heart catheterization, and results emanating from this subgroup should be considered as hypothesis-generating. Finally, the delay between initial HF diagnosis and CS occurrence is not available in our database and may carry relevant information.

## 6. Conclusions

In summary, our results emphasize the great heterogeneity among patients presenting with CS. The presence or absence of a pre-existing HF is a significant part of this heterogeneity. CS patients of the worsening CHF group were older and had lower LVEFs, more extra-cardiac comorbidities, more congestion, and more organ failure as compared to those of the new-onset HF group. Importantly, this translated in a higher risk profile and a 30% increase in in-hospital all-cause mortality, although this association disappeared after adjustment. The heterogeneity of this population prompts us to better determine the phenotypes of patients in terms of clinical, biological, and hemodynamic characteristics, but also HF duration, which may have important implications for future trials and evaluations of tailored therapies.

## Figures and Tables

**Figure 1 life-12-01844-f001:**
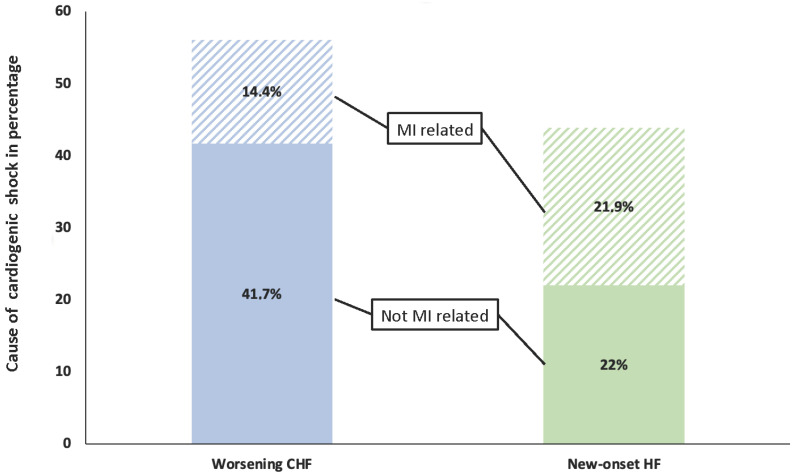
Graphic representation of the cardiogenic shock population based on the presence or absence of previous history of heart failure and the presence or absence of an acute myocardial infarction at admission. CS = cardiogenic shock, MI = myocardial infarction, HF = heart failure, CHF = chronic heart failure.

**Figure 2 life-12-01844-f002:**
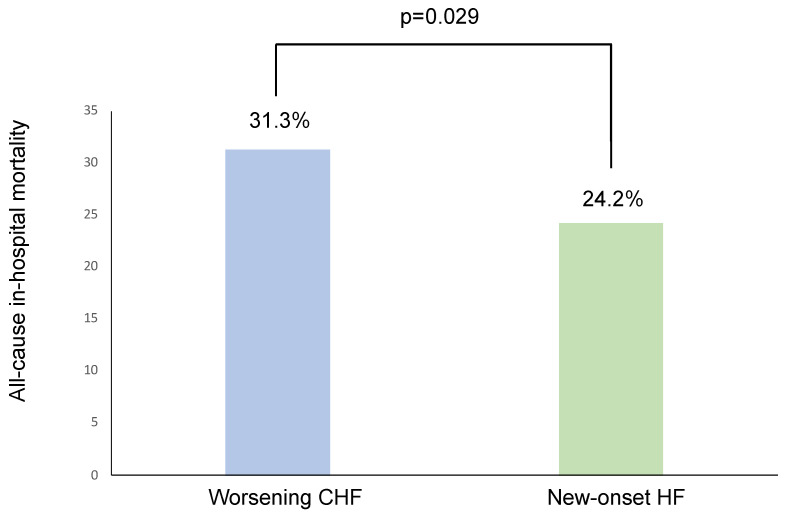
All-cause in-hospital mortality in both groups. HF = heart failure, CHF = chronic heart failure.

**Table 1 life-12-01844-t001:** Baseline characteristics.

	Overall Population	New-Onset HF	Worsening CHF	*p* Value
	n = 772	n = 339	n = 433	
Sex male	552 (71.5)	222 (65.5)	330 (76.2)	0.001
Age (years)	65.7 (14.9)	62.7 (16.2)	68 (13.4)	<0.001
Diabetes	217 (28)	73 (21.5)	144 (33.3)	<0.001
Hypertension	364 (47.2)	136 (40.1)	228 (52.7)	0.001
Active smokers	206 (27.8)	112 (34.8)	94 (22.4)	<0.001
Hypercholesterolemia	277 (35.9)	76 (22.4)	201 (46.4)	<0.001
BMI (Kg/m^2^)	25.8 (5.5)	25.8 (5.3)	25.8 (25.7)	0.995
COPD or chronic respiratory failure	54 (7)	16 (4.7)	38 (8.8)	0.029
Renal failure (Cl < 60 mL/min)	164 (21.2)	24 (7.1)	140 (32.3)	<0.001
Peripheral artery disease	114 (14.8)	24 (7.1)	90 (20.7)	<0.001
Treatment before admission				
Aspirin	288 (37.4)	88 (26.1)	200 (46.2)	<0.001
P2Y12 inhibitors	126 (16.3)	53 (15.7)	73 (16.8)	0.671
Anticoagulants	221 (28.7)	34 (10.1)	187 (43.2)	<0.001
VKA	165 (21.4)	24 (7.1)	141 (32.6)	<0.001
DOA	56 (7.2)	10 (3)	46 (10.6)	<0.001
Betablockers	316 (41)	67 (19.9)	249 (57.5)	<0.001
ACEI or ARB	292 (37.9)	70 (20.8)	222 (51.3)	<0.001
Sacubitril/Valsartan	18 (2.3)	0	18 (4.4)	0.001
Aldosterone antagonists	108 (14)	9 (2.7)	99 (22.9)	<0.001
Loop diuretics	376 (48.7)	65 (19.3)	311 (71.8)	<0.001
Statins	286 (37)	66 (19.6)	220 (50.8)	<0.001

In this table, continuous variables are expressed as mean +/− SD. HF = heart failure, CHF = chronic heart failure, BMI = body mass index, SD = standard deviation, COPD = chronic obstructive pulmonary disease, VKA = vitamin K antagonist, DOA = direct oral anticoagulant, ACEI or ARB = angiotensin-converting enzyme inhibitor and an angiotensin receptor blocker.

**Table 2 life-12-01844-t002:** Clinical presentation at admission.

	Overall Population	New-Onset HF	Worsening CHF	*p* Value
	n = 772	n = 339	n = 433	
Heart Rate (bpm)	95.6 (29.6)	100 (31.1)	91 (27.7)	<0.001
Systolic blood pressure (mmHg)	101 (25.2)	102 (25.6)	100 (24.8)	0.301
Diastolic blood pressure (mmHg)	63.2 (17.4)	65.2 (18.2)	61 (16.6)	0.005
Mean blood pressure (mmHg)	74.9 (18.3)	76.6 (18.8)	73.6 (17.9)	0.023
Sinus rhythm	399 (51.9)	199 (59.4)	200 (46.2)	<0.001
LVEF (%)	26.3 (13.4)	27.9 (14.5)	25 (12.2)	0.003
LVEF ≤ 30%	551 (72.2)	228 (68.1)	323 (75.5)	0.023
Precipitating factor None	115 (14.9)	31 (9.1)	84 (19.4)	<0.001
Sepsis	92 (11.9)	33 (9.7)	59 (13.6)	0.024
Myocardial infarction	280 (36.3)	169 (49.9)	111 (25.6)	<0.001

In this table, continuous variables are expressed as mean +/− SD. HF = heart failure, CHF = chronic heart failure, BMI = body mass index, SD = standard deviation, LVEF = left ventricle ejection fraction.

**Table 3 life-12-01844-t003:** Biology parameters at admission.

	Overall Population	New-Onset HF	Worsening CHF	*p* Value
	n = 772	n = 339	n = 433	
eGFR (mL/min) (n = 761)	46 [28–67]	56 [39–77]	40 [25–58]	<0.001
Creatinine (mmol/L) (n = 751)	133 [95–190]	110 [84–149]	145 [112–210]	<0.001
Hemoglobin (g/dL) (n = 754)	12.6 [11–14]	13 [11.2–14.6]	12 [10.6–14]	<0.001
Arterial blood lactate (mmol/L) (n = 684)	3 [2–4.75]	3 [2–5]	2.9 [2–4]	0.121
Arterial blood lactate > 2.2 (mmol/L) (n = 684)	422 (61.7)	200 (65.8)	222 (58.4)	0.049
pH (n = 668)	7.39 [7.28–7.46]	7.37 [7.26–7.44]	7.40 [7.30–7.46]	0.007
PT (%) (n = 731)	59 [37–77]	70 [52–85]	48 [29–68]	<0.001
SGOT (IU/L) (n = 547)	90 [39–301]	125 [51–377]	69 [35–220]	0.521
SGPT (IU/L) (n = 559)	59 [27–183]	71 [33–194]	48 [24–177]	0.815
Bilirubin (mmol/L) (n = 544)	16 [9–29]	13 [8–23]	20 [11–34]	<0.001
BNP (ng/L) (n = 264)	1150 [476–2778]	835 [277–2051]	1511 [687–3157]	0.182
NT-pro-BNP (ng/L) (n = 224)	9277 [4045–23,810]	6306 [2063–11,730]	12,652 [5360–30,000]	0.006
BNP or NT-pro-BNP by deciles (n = 480)				
1	47 (9.8)	36 (18.8)	11 (3.8)	
2	49 (10.2)	27 (14)	22 (7.6)	
3	51 (10.6)	24 (12.5)	27 (9.4)	
4	45 (9.3)	18 (9.4)	27 (9.4)	
5	47 (9.8)	21 (10.9)	26 (9)	<0.001
6	52 (10.8)	14 (7.2)	38 (13.2)	
7	46 (9.5)	13 (6.8)	33 (11.5)	
8	50 (10.4)	14 (7.2)	36 (12.5)	
9	41 (8.5)	9 (4.7)	32 (11.1)	
10	52 (10.8)	16 (8.3)	36 (12.5)	
CRP (mg/L) (n = 406)	28 [9–69]	34 [8–98]	26 [10–58]	0.006

In this table, continuous variables are expressed as median [IQR]. HF = heart failure, CHF = chronic heart failure, eGFR = estimated glomerular filtration rate, IQR = interquartile range, PT = prothrombin time, SGOT = Serum Glutamooxaloacetate Transferase, SGPT = Serum Glutamopyruvate Transferase, CRP = C-reactive protein, BNP = Brain natriuretic peptide.

**Table 4 life-12-01844-t004:** Right heart catheterization parameters.

	Overall Population	New-Onset HF	Worsening CHF	*p* Value
	n = 83	n = 36	n = 47	
Right atrial pressure (mmHg)	10 [6–14]	8 [5–11]	12 [6–16]	0.028
Mean pulmonary arterial pressure (mmHg)	29 [24–35]	28 [21–33]	32 [26–38]	0.157
Pulmonary capillary wedge pressure (mmHg)	19 [14–25]	16 [13–23]	22 [15–29]	0.062
Cardiac index (L/min/m^2^)	2.1 [1.9–3]	2.1 [1.9–3]	2 [1.8–2.8]	0.391

In this table, continuous variables are expressed as median [IQR]. HF = heart failure, CHF = chronic heart failure, IQR = interquartile range.

**Table 5 life-12-01844-t005:** Shock management.

	Overall Population	New-Onset HF	Worsening CHF	*p* Value
	n = 772	n = 339	n = 433	
ICU length of stay (days)	11 [7–21]	10 [6–20]	12 [8–21]	0.095
In-hospital length of stay (days)	16 [10–26]	14 [9–24]	16 [10–27]	0.056
Intravenous diuretics	633 (82)	259 (76.9)	374 (86.8)	<0.001
Volume expansion	321 (41.6)	168 (49.9)	153 (35.6)	<0.001
Dobutamine	632 (82.2)	274 (81.3)	358 (83)	0.534
Duration (days)	5 [2–8]	4 [2–6]	5 [3–9]	0.027
Max dose < 10 gamma/Kg/min	405 (68.9)	179 (70)	226 (68.1)	
Max dose 10–15 gamma/Kg/min	136 (23.1)	62 (24.2)	74 (22.3)	0.241
Max dose > 15 gamma/Kg/min	47 (7.9)	15 (5.9)	32 (9.6)	
Noradrenaline	410 (53.3)	195 (57.9)	215 (49.9)	0.037
Duration (days)	3 [2–6]	3 [2–5]	3 [2–6]	0.134
Max dose < 1 mg/h	86 (22.8)	42 (23.1)	44 (22.6)	
Max dose 1–5 mg/h	215 (57)	98 (53.8)	117 (60.3)	0.309
Max dose > 5 mg/h	75 (19.9)	42 (23.1)	33 (17)	
Adrenaline	95 (12.3)	51 (15.1)	44 (10.2)	0.042
Duration (days)	1 [1–2]	1 [1–2]	1 [1–4]	0.407
Max dose < 1 mg/h	34 (38.6)	16 (34)	18 (43.9)	
Max dose 1–5 mg/h	40 (45.4)	22 (46)	18 (43.9)	0.533
Max dose > 5 mg/h	14 (15.9)	9 (19.1)	5 (12.2)	
Levosimendan	57 (7.4)	19 (5.6)	38 (8.8)	0.092
Renal replacement therapy	122 (15.8)	52 (15.3)	70 (16.2)	0.745
Mechanical ventilation	291 (37.9)	175 (51.8)	116 (26.9)	<0.001
Mechanical circulatory support	143 (18.6)	87 (25.6)	56 (13)	<0.001
IABP	48 (34.5)	31 (36.5)	17 (31.5)	0.552
Duration (days)	2 [1–5]	2 [1–3]	4 [2–5]	0.942
ECLS	85 (60.7)	55 (64)	30 (55.6)	0.326
Duration (days)	5 [3–8]	5 [2–9]	7 [3–8]	0.665
Impella	26 (18.7)	19 (22.4)	7 (12.9)	0.178
Duration (days)	6 [3–9]	8 [4–10]	5 [1–7]	0.213
Coronary angiography	399 (51.7)	233 (68.7)	166 (38.3)	<0.001
At least one vessel disease	321 (41.6)	185 (54.5)	136 (31.4)	0.007
PCI of culprit lesion	256 (33.2)	171 (50.4)	85 (19.6)	<0.001

In this table, continuous variables are expressed as median [IQR]. HF = heart failure, CHF = chronic heart failure, ICU = intensive care unit, IQR = interquartile range, IABP = intra-aortic balloon pump, ECLS = extracorporeal life support, PCI = percutaneous coronary intervention.

**Table 6 life-12-01844-t006:** Multivariate analysis (in-hospital all-cause mortality).

	Odds Ratio	95% Confidence Interval	*p* Value
Sex male	1.149	0.737–1.794	0.540
Age (per year)	1.028	1.012–1.043	<0.001
Myocardial infarction as precipitating factor	1.451	0.949–2.220	0.086
Heart rate (per 1 bpm)	1.008	1.001–1.014	0.019
Systolic blood pressure (per 1 mmHg)	0.987	0.979–0.995	0.002
LVEF (per 1%)	0.975	0.959–0.992	0.003
Creatinin (per 1 mmol/L)	1.002	0.999–1.004	0.147
Hemoglobin (per 1 g/dL)	0.901	0.822–0.987	0.026
Arterial blood lactate (per 1 mmol/L)	1.101	1.042–1.163	0.001
PT (per 1%)	0.992	0.983–1.001	0.070
Worsening CHF	1.051	0.680–1.624	0.823

LVEF = left ventricle ejection fraction, CHF = chronic heart failure, PT = prothrombin time.

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
