# Peer review of "Impact of Pre-Existing History of Heart Failure on Patient Profile, Therapeutic Management, and Prognosis in Cardiogenic Shock: Insights from the FRENSHOCK Registry"

_life, 2022, doi:10.3390/life12111844_

Round 1

Reviewer 1 Report

Daear Sirs,

I read with great interest the manuscript entitled:

„Impact of pre-existing history of heart failure on patient profile, 2 therapeutic management and prognosis in cardiogenic shock: 3 Insights from the FRENSHOCK registry”

I received for review an interesting, well-written manuscript on the care of patients with cardiogenic shock. The still high mortality rate in this group of patients calls for more knowledge on the subject. This was attempted by the authors of this manuscript. The analysis has the character of a prospective registry, including a rather large but heterogeneous group of patients. And only in-hospital parameters were analysed. However, given the difficulties in setting up randomised trials on this group of patients, it enriches our knowledge on this topic.

Having examined the document submitted, I have a few comments:

1.     There is quite a high percentage of men included in the work - such a population, is there a factor that influences this - this could possibly be taken up in discussion.

2.     No information is available on sodium-glucose cotransporter 2 (SGLT2) inhibitor therapy. We have evidence that this group of drugs significantly improves prognosis and reduces the risk of re-admission.

3.     I leave it to the discretion of the authors to perform the analysis according to the aetiology of shock - e.g. based on the aetiology cited in the study; or to include different aetiologies as variables in a multivariate analysis; the absence of differences could lend credence to the proposition that it is the patient's haemodynamic-metabolic state, rather than the aetiology, that is associated with in-hospital prognosis. This could allow for clearer conclusions to be drawn from the prospective observation carried out.

4.     It seems to me that with a fairly high rate of need for PCI (PCI of culprit lesion in 50.4% of patients with CS in de novo HF), the high rate of use of IABP in light of current recommendations and the results of the IABP-SHOCK trial - may require commentary.

5.     I leave this to the discretion of the authors. In the discussion, the authors refer to a paper from which CS patients were excluded. Whether this is certainly justified?

6.     I leave it to the authors to consider citing a paper [J. Clin. Med. 2021, 10, 5059; https://doi.org/10.3390/jcm10215059] that analysed more than 10,000 patients with CS-induced MI - while looking at baseline characteristics.

Sincerely,

Reviewer

Author Response

Thank you very much for your interest in our work.

  1. There is quite a high percentage of men included in the work - such a population, is there a factor that influences this - this could possibly be taken up in discussion.

Thank you for this comment.

Our results are in perfect line with the previous literature in the field. There were 28.5% of women in our study. As an example, among patients included in CULPRIT-SHOCK and IABP-SHOCK II trials, 24% and 31.2% were women (very similar to our analysis), respectively. Overall, these previous studies showed that despite apparent worse clinical profile, mortality between male and female was similar. As well, in our study, sex was not associated with outcomes after multivariate analysis (as shown in Table 6). Therefore, if editor agrees, we prefer to not discuss further this specific point in order to not unnecessarily lengthen the manuscript.

  1. No information is available on sodium-glucose cotransporter 2 (SGLT2) inhibitor therapy. We have evidence that this group of drugs significantly improves prognosis and reduces the risk of re-admission.

Thank you for this comment.

We perfectly agree with this comment, especially after the results of several large studies that clearly established the role of SGLT2 inhibitors in heart failure. However, these studies were published after 2019. The FRENSHOCK registry was a national French multicentre registry conducted between April and October 2016. At that time, SGLT2 inhibitors were not yet recommended for patients with HF and not available in France.

  1. I leave it to the discretion of the authors to perform the analysis according to the aetiology of shock - e.g. based on the aetiology cited in the study; or to include different aetiologies as variables in a multivariate analysis; the absence of differences could lend credence to the proposition that it is the patient's haemodynamic-metabolic state, rather than the aetiology, that is associated with in-hospital prognosis. This could allow for clearer conclusions to be drawn from the prospective observation carried out.

Thank you for your comment.

As discussed in the first paragraph of the discussion (Page 9 and 10), aetiology of the pre-existing cardiopathy is different from the aetiology of the index cardiogenic shock (precipitating factor), although both could be linked.

Our analysis already takes into account the aetiology of shock (precipitating factor). Indeed, “MI as a precipitating factor” has been included in the multivariate analysis (see Table 6) and was not independently associated with outcomes in our analysis. We agree that the absence of independent association between this variable and outcome indeed lends credence to the proposition that it is the patient's haemodynamic-metabolic state, rather than the aetiology of shock that is associated with prognosis.

Besides, doing an analysis based on aetiologies of the pre-existing cardiopathy may lead to very small groups (6 groups, all of less than 100 patients) and this is the reason why we decided to not do it. In addition, 85 (11%) patients had mixed causes of the pre-existing cardiopathy.

  1. It seems to me that with a fairly high rate of need for PCI (PCI of culprit lesion in 50.4% of patients with CS in de novo HF), the high rate of use of IABP in light of current recommendations and the results of the IABP-SHOCK trial - may require commentary.

Thank you for this observation.

Despite recommendations and results of randomized trials, IABP is still widely use in clinical practice when no other hemodynamic support is accessible (Helleu B, Auffret V, Bedossa M, et al. Current indications for the intra-aortic balloon pump: the CP-GARO registry. Arch Car- diovasc Dis 2018;111:739—48 ; Strom JB, Zhao Y, Shen C, et al. National trends, predictors of use, and in-hospital outcomes in mechanical circulatory sup- port for cardiogenic shock. EuroIntervention 2018;13:e2152—9). Advanced mechanical circulatory support possibilities are indeed not available in all cardiac intensive care units (levels 2 and 3 ICUs according to the Acute Cardiovascular Care Association) and real-life data like our registry reveals us that a high proportion of cardiogenic shock patients initially present to less-well-resourced ICUs. This observation might explain the high rate of IABP use in the FRENSHOCK study.

The limitation section (Page 11) has been modified to address this comment.

  1. I leave this to the discretion of the authors. In the discussion, the authors refer to a paper from which CS patients were excluded. Whether this is certainly justified?

Thank you for your comment.

We agree that some references do not specifically deal with cardiogenic shock but with heart failure. We however feel that some results observed in patients with heart failure are important to know to better apprehend our own results. Therefore, if you agree, we would like to keep the manuscript unchanged regarding this point.

  1. I leave it to the authors to consider citing a paper [J. Clin. Med. 2021, 10, 5059; https://doi.org/10.3390/jcm10215059] that analysed more than 10,000 patients with CS-induced MI - while looking at baseline characteristics.

As requested, the citation has been added to the manuscript in the discussion section (new ref # 17). All numbers of references have been updated accordingly.

Reviewer 2 Report

This study confirms that older patients with faster heart rate and lower BP and lower EF who are anemic and have higher lactates do poorly.

I would have expected the acute CS patients to do poorly since they did not have time to compensate for the acute changes; how can you explain that.

Author Response

This study confirms that older patients with faster heart rate and lower BP and lower EF who are anemic and have higher lactates do poorly.

I would have expected the acute CS patients to do poorly since they did not have time to compensate for the acute changes; how can you explain that.

Thank you very much for your comment and your interest in our work. We perfectly agree with this comment. Our discussion (Page 11) is indeed in perfect accordance with it.

In our analysis new-onset HF patients (acute CS) had much lesser co-morbidities than worsening HF ones and this may have counter balanced the results in our study and may be the main reason why worsening HF patients did worse in univariate analysis. The difference in mortality rates completely disappeared between groups after multivariate analysis further highlighting this point.

Reviewer 3 Report

Schurtz and co-workers compared cardiogenic shock patients with new-onset heart failure to those with worsening CHF in FRENSHOCK registry and revealed the great heterogeneity of the patients presenting with cardiogenic shock. Despite the great heterogeneity, the history of CHF was not the significant risk factor of in-hospital mortality by multivariate logistic regression model.

“The great heterogeneity in the background of patients who suffer from cardiogenic shock” is the only message that could be learned from this study, and this is not especially new information. On the other side, the manuscript is well organized and well written.

Author Response

Thank you very much for your comment and your interest in our work

Reviewer 4 Report

This manuscript presents new findings in the multicentre FRENSHOCK registry by comparing cardiogenic shock patients with newly developed heart failure with those with worsening chronic heart failure. The authors found that patients with worsening chronic heart failure experienced higher in-hospital all-cause mortality and had a higher risk profile. They conclude that this appears to have led to a 30% increase in in-hospital all-cause mortality.

The manuscript is essentially well written and much of the data supports the authors' conclusions. However, it would be better if a few areas of concern could be improved.

1) The first line of Table 1 mentions Sex, but this does not make it clear whether it is male or female.

2) The entry in Table 2 says Mean +/- SD (%), which makes the table difficult to read. It would be easier to understand if it were written at the bottom of the Table.

3) The pie chart in Figure 1 is difficult to read. There is no need to make it a pie chart, just show the MI involvement in Worsening CHF and New onset HF, respectively, as a percentage.

4) Figure 2. all-cause in-hospital mortality in both groups, but would like to know if there is a significant difference. p-values should be stated.

It is well written and I would accept it if the above minor is corrected.

Author Response

Thank you for your interest in our work

  1. The first line of Table 1 mentions Sex, but this does not make it clear whether it is male or female.

Thank you for this comment. As requested, we have modified the Table 1 to make it clearer for readers.

  1. The entry in Table 2 says Mean +/- SD (%), which makes the table difficult to read. It would be easier to understand if it were written at the bottom of the Table.

Thank you for this comment. As requested, we have modified the Table 2 to make it clearer for readers. We have also modified other tables accordingly to keep them homogeneous.

  1. The pie chart in Figure 1 is difficult to read. There is no need to make it a pie chart, just show the MI involvement in Worsening CHF and New onset HF, respectively, as a percentage.

As requested, the Figure 1 has been completely modified to make it easier to read.

  1. Figure 2. all-cause in-hospital mortality in both groups, but would like to know if there is a significant difference. p-values should be stated.

As requested, the Figure 2 has been modified and we have included p values.

Round 2

Reviewer 1 Report

I approve the manuscript as submitted. Thank you to the authors for your response.